

# Recovery of sweet taste preference in adult rats following bilateral chorda tympani nerve transection

Andrew Padalhin[1], Celine Abueva[1,2], So Young Park[1], Hyun Seok Ryu[3], Hayoung Lee[3], Jae Il Kim[4], Phil-Sang Chung[1,2,5] and Seung Hoon Woo[1,2,5]

[1] Beckman Laser Institute Korea, College of Medicine, Dankook University, Cheonan, Chungcheongnam-do, Republic of Korea
[2] Medical Laser Research Center, Dankook University, Cheonan, Chungcheongnam-do, Republic of Korea
[3] Interdisciplinary Program for Medical Laser, College of Medicine, Dankook University, Cheonan, Chungcheongnam-do, Republic of Korea
[4] Department of Neurology, Dankook University College of Medicine, Dankook University Hospital, Cheonan, Chungcheongnam-do, Republic of Korea
[5] Department of Otorhinolaryngology-Head and Neck Surgery, Dankook University College of Medicine, Cheonan, Chungcheonam-do, Republic of Korea

Corresponding author
Seung Hoon Woo,
lesaby@hanmail.net

## ABSTRACT

**Background:** Numerous studies have noted the effect of chorda tympani (CT) nerve transection on taste sensitivity yet very few have directly observed its effects on taste receptor and taste signaling protein expressions in the tongue tissue.

**Methods:** In this study, bilateral CT nerve transection was performed in adult Sprague Dawley rats after establishing behavioral taste preference for sweet, bitter, and salty taste *via* short term two-bottle preference testing using a lickometer setup. Taste preference for all animals were subsequently monitored. The behavioral testing was paired with tissue sampling and protein expression analysis. Paired groups of CT nerve transected animals (CTX) and sham operated animals (SHAM) were sacrificed 7, 14, and 28 days post operation.

**Results:** Immunofluorescence staining of extracted tongue tissues shows that CT nerve transection resulted in micro-anatomical changes akin to previous investigations. Among the three taste qualities tested, only the preference for sweet taste was drastically affected. Subsequent results of the short-term two-bottle preference test indicated recovery of sweet taste preference over the course of 28 days. This recovery could possibly be due to maintenance of T1R3, GNAT3, and TRPM5 proteins allowing adaptable recovery of sweet taste preference despite down-regulation of both T1R2 and Sonic hedgehog proteins in CTX animals. This study is the first known attempt to correlate the disruption in taste preference with the altered expression of taste receptors and taste signaling proteins in the tongue brought about by CT nerve transection.

## INTRODUCTION

Ingestion of materials as food is regulated through orosensory stimulation such as the olfactory and gustatory systems. These afferent systems are composed of sensory organs, innervation, and associated brain processes. The binding of exogenous chemicals to specific taste-signaling proteins in the tongue results in the detection of taste—a chemosensory event. This is made possible *via* the activation of certain ion channels and resulting action potential for signal transduction, particularly through chorda tympani, a nerve that has been long identified for this purpose. Upon reaching the brain, these signals are processed primarily for identification of the taste quality, whether it's salty, sweet, sour, or bitter. In addition to the identification of the specific taste (or combination of which), the relative concentration is also simultaneously determined. The combined analysis leads to either ingestion of nourishment based on associating preferable taste qualities on certain items or rejection of potentially toxic chemicals *via* association of aversive taste qualities.

The chorda tympani (CT) is the main peripheral nerve relaying taste signals from the anterior two-thirds of the tongue towards the nucleus of the solitary tract (NST). It is considered a branch of the facial nerve and shares NST terminal field innervation with the greater superficial petrosal and glossopharyngeal nerve. Taste disturbance and changes in taste sensitivities have been attributed mainly to CT nerve injury (*Guinand et al., 2010*; *Mueller et al., 2008*; *Cain, Frank & Barry, 1996*; *Bonardi et al., 2016*; *Pittman et al., 2007*; *Stratford, Curtis & Contreras, 2006*). Some studies have been conducted regarding elucidating the effect of chorda tympani manipulation on anatomical composition and taste detection. Among the notable findings on the effect of CT transection is its effect on taste bud volume and papillae morphology. Unilateral CT transection has been found to induce size reduction of the taste buds on the ipsilateral side and hyperplasia of taste buds on the contralateral side, implying anatomical change counteracting the effect of denervation (*Li et al., 2015*; *Sollars, 2005*; *Reddaway et al., 2012*). Further investigation by *Sollars & Bernstein (2000)* elucidated that the chorda tympani displays age-related plasticity, making it markedly vulnerable when damaged during early development in young rats compared to adults. In addition to anatomical changes resulting from denervation, transection of the chorda tympani has also been effective in inducing changes in taste perception in animals. A follow-up study concerning neonatal CT transection showed that the peripheral taste system in adult rodents is capable of functional recovery for salt taste detection compared to neonatal rats that underwent the same procedure (*Martin & Sollars, 2015*; *Kopka, Geran & Spector, 2000*). Other studies have also noted that CT transection prompted modified taste sensitivity to free fatty acids such as linoleic and oleic acids (*Pittman et al., 2007*; *Stratford, Curtis & Contreras, 2006*).

The effect of denervation on the terminal field volume of the different nerves connecting to the NST has also been investigated concerning CT regeneration and plasticity. Transection of greater superficial petrosal and glossopharyngeal nerves resulted to increase field terminal volume by CT possibly for compensatory effect from the loss of afferent input from severed nerves (*Corson & Hill, 2011*). Conversely, regenerative failure following CT transection results in terminal field plasticity of the remaining intact glossopharyngeal

and greater superficial petrosal nerve (*Martin et al., 2019*), and that brain-derived neurotrophic factor (BDNF) is crucial in regenerating damaged CT nerves (*Meng et al., 2017*). In one study, the pattern of CT nerve innervation in the NST has also been found to be influenced by the dietary salt content in rats and indicating its possible effect on behavioral study outcomes when testing animals with altered salt intake during the early developmental stage (*Pittman & Contreras, 2002*). So far the majority of the studies involved in establishing how CT nerve transection centered on observing some behavioral changes in taste sensitivity, terminal field innervation of the NST, and morphological transformation of the taste buds and papillae.

Though it has been stated that innervation is a crucial factor in the proper development of taste buds and tongue papillae (*Reddaway et al., 2012*), limited information has been provided concerning its effect on the maintenance and functionality of taste receptor proteins in the taste cells. There is currently limited publication relating the consequence of CT nerve transection on the expression of certain taste receptors and the observable change in behavioral taste preference. This study focuses on validating the behavioral change following CT nerve transection *via* observing relative changes in taste protein receptors present in remaining taste buds up to 4 weeks after the procedure. Observing changes in taste receptor protein expression in the remaining taste buds could provide information on how taste preference is modified through CT denervation.

## MATERIALS AND METHODS

### Behavioral testing for taste preference

In this study, Sprague Dawley rats were used to determine the effect of chorda tympani transection on the taste preference behavior and with respect to protein expression of some taste receptors and transduction proteins considering it is a well-known animal model for not only biomedical studies but also for behavioral studies. (*Sollars, 2005*; *Golden et al., 2011*; *Spector & Grill, 1992*; *Tordoff, Alarcon & Lawler, 2008*) The *in vivo* experiments were performed in accordance with the guidelines set by the Institutional Animal Care and Use Committee at Dankook University (DKU-20-036). All experimenters were aware of the group allocation during the different stages of the study conducted with non-registered protocols. The animal number was determined based on the capacity of the lickometer setup that will accommodate the simultaneous testing of several animals per day. For this study 36 male Sprague-Dawley rats, 7 weeks of age and approximately 250 grams in weight purchased from Orientbio Co., Seongnam, South Korea. Upon arrival in the research facility, the animals were initially divided into 12 cages, housing three rats per cage. The rats were allowed to acclimatize for 3 days in a temperature-controlled animal room with a 12-h timed light-dark cycle and provided with food and water *ad libitum*. The caged rats were then divided into two categories and three observation groups. Half of the animals underwent bilateral chorda tympani transection and were categorized as the CTX group while the remaining half underwent a sham operation and were categorized as the SHAM (control) group for comparison with the CTX group. Sampling and observation periods were set 7, 14, and 28 days post-chorda tympani transection with allotted six CTX rats and six SHAM rats per observation period. Figure 1 shows the schematic diagram of

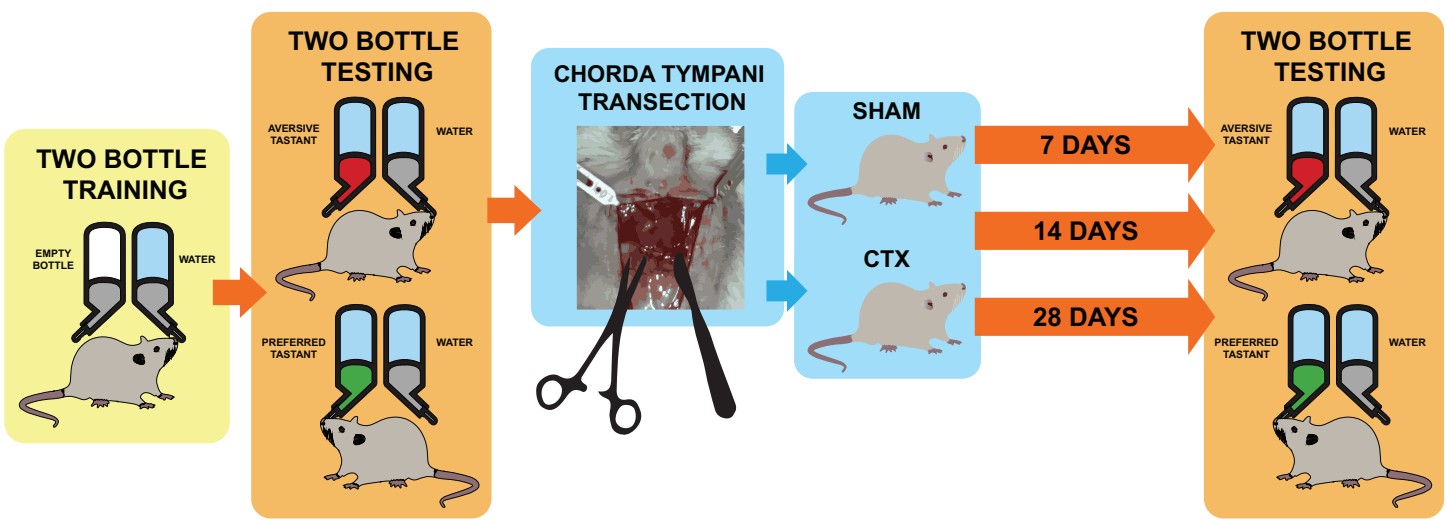

**Figure 1 Schematic diagram for taste behavior testing before and after chorda tympani transection.** Schematic diagram for taste behavior testing. Prior to the bilateral transection of the chorda tympani nerve, all animals are acclimatized and trained using a two-bottle setup for 1 week. Baseline taste preference and aversion is established using three taste qualities: sweet (sucrose solution-preferred), salty (NaCl solution-avoided), and bitter (denatonium solution-avoided). Upon completing the baseline reading, animals undergo either sham or bilateral chorda tympani transection. After recovery, the taste preference and aversion is measured again after 7, 14, and 24 days post-transection using the same randomized two-bottle testing method.                         

how the study was carried out. Detailed chorda tympani transection procedure will be discussed in the succeeding section. The taste preferences of the animals were determined using a triple lickometer setup (Model 80380, Lafayette Instrument, Sagamore, IND, USA). Three tastant solutions were formulated based on the lowest concentrations stated in a previously published reference detailing the preference of certain rat strains to different tastant (*Tordoff, Alarcon & Lawler, 2008*). As per reference, the sweet taste was prepared at its lowest preferred concentration by dissolving sucrose (Cat#.S9378, Sigma-Aldrich, St. Louis, MO, USA) in distilled water to create a 10 mM solution. The salty taste was formulated at its lowest aversive concentration of 562 mM by dissolving NaCl (Cat#.7548-4400, Daejung, Korea) in distilled water. Lastly, a tastant solution for bitter taste was made by dissolving denatonium benzoate (Cat#.30914, Sigma-Aldrich, St. Louis, MO, USA) in distilled water at its lowest aversive concentration of 0.0316 mM. To establish a baseline reading of the taste preferences, all animals were subjected to a modified two-bottle preference test with a tastant at a specified concentration and a known response with triple distilled water as a control. The detailed preference test schedule is listed in Table 1. Testing preference for both salty and bitter taste was conducted under food and water-deprived conditions to motivate the animals to drink from the bottles while the test for sweet taste was carried out under normal conditions to establish partiality of the animals to said taste quality over simple distilled water. Scheduled food and water depravation were devised in reference to previous studies dealing with the interaction of hunger and thirst (*Oatley & Tonge, 1969*; *Bolles, 1961*; *Finger & Reid, 1952*), the effect of aging to taste sensitivity (*Inui-Yamamoto et al., 2017*), and to maximize our ability to measure animals' response within the capacity of the available equipment. Hunger
**Table 1 Schedule of two bottle training and testing for determining taste preference and aversion in rats.**

| DAYS | Training/Test condition | | Animal condition during testing |
|---|---|---|---|
| 3 | Acclimatization | • Food and water provided *ad libitum* | Non–deprived |
| 1 | Two Bottle Training (condition animals to drink from two bottles) | • Rats are transferred into the Lickometer setup<br>• Test is run for 1 h per group.<br>• An empty bottle and one water filled bottle is provided, position is switched every 15 min<br>• Food and water is removed overnight | Non–deprived |
| 1 | Two Bottle Training (condition animals to drink from two bottles) | • Rats are transferred into the Lickometer setup<br>• Test is run for 1 h per group.<br>• An empty bottle and one water filled bottle is provided, position is switched every 15 min<br>• Food and water is removed overnight | Deprived |
| 1 | Two Bottle Training (condition animals to drink from two bottles) | • Rats are transferred into the Lickometer setup<br>• Test is run for 1 h per group<br>• An empty bottle and one water filled bottle is provided, position is switched every 15 min<br>• Food and water is provided after testing | Deprived |
| 1 | Two Bottle Training (condition animals to drink from two bottles) | • Rats are transferred into the Lickometer setup<br>• Test is run for 1 h per group<br>• An empty bottle and one water filled bottle is provided, position is switched every 15 min<br>• Food and water is provided after testing | Deprived |
| 2 | Rest | • Food and water provided *ad libitum* | Non–deprived |
| 1 | Two Bottle preference test (Preferred taste quality) | • Sweet taste preference testing<br>• Test is run for 1 h per group<br>• Sucrose solution is provided as tastant<br>• One bottle with tastant and one bottle with water is provided, position is switched every 15 min<br>• Food and water is removed overnight | Non–deprived |
| 1 | Two Bottle preference test (Aversive taste quality) | • Bitter taste preference testing<br>• Denatonium solution is provided as tastant<br>• One bottle with tastant and 1 bottle with water is provided, position is switched every 15 min<br>• Food and water is removed overnight | Deprived |
| 1 | Two Bottle preference test (Aversive taste quality) | • Salty taste preference testing<br>• NaCl solution is provided as tastant<br>• One bottle with tastant and one bottle with water is provided, position is switched every 15 min<br>• Food and water is provided after testing | Deprived |
| 1 | Rest | • Food and water provided *ad libitum* | Non–deprived |
potentiated thirst motivating the animal to drink more than the usual amount to replenish the water deficit (*Oatley & Tonge, 1969*). However prolonged hunger eventually induces self-imposed restrictions to both food and water intake (*Bolles, 1961*; *Finger & Reid, 1952*), thus brief access to food was provided during the testing period to mitigate this effect during subsequent testing of aversive tastants. All animals underwent one trial of taste preference testing per taste quality on separate days. Each trial consisted of four rounds with an allotted time of 15 min each. Bottles containing either the tastant or distilled water were placed in the first and last lickometer slot and for each test round, the position of each bottle was switched. Food pellets were provided at the third round of each trial to further motivate the animal in drinking from the test bottles. To eliminate the possible effect of conditioned response due to routine lickometer testing, animals were not tested in the same cage during each behavioral test. Bottles of the tastant and water were also switched in each setup. After the test, the animals were returned to their housing cage and had an additional 1 h to drink before subsequent food and water deprivation in preparation for next-day testing (for aversive tastants). The total number of licks on both the tastant bottle and the water bottle was recorded using the Scurry Activity Monitoring Software (Model 86165 Lafayette Instrument, Sagamore, IN, USA). Taste preference was calculated based on the following formula:

$$\text{Preference \%} = \left( \frac{N_t}{N_t + N_w} \right) 100\%$$

where: $N_t$ = number of lick on tastant bottle
$N_w$ = number of licks on water bottle.

## Bilateral chorda tympani transection

Upon completion of the preliminary test establishing behavioral taste preference, each cage was randomly selected for either chorda tympani transection or sham operation (control group). As previously stated, the animal subjects were categorized into either the CTX or the SHAM group and divided based on tissue sampling periods. Eighteen rats underwent bilateral chorda tympani transection (CTX) while the remaining eighteen were sham-operated (SHAM). For this procedure, the animals were weighed and anesthetized *via* intramuscular injection of Zoletil (15 mg/kg, VIRBAC) and Rompun (5 mg/kg, BAYER) at a ratio of 3:2. Upon complete anesthetic induction, fur was removed by shaving and subsequent application of hair removal cream on the ventral region of the neck. The bare skin was then cleaned with 70% ethanol and disinfected with a povidone-iodine solution. A 1.5 cm full-thickness skin incision was made along the midline of the neck to expose the underlying tissues. The ventromedial portion of both masseter muscles, flanking the centrally located anterior digastric muscle, are identified and used as landmarks for approaching the space between the inner side of the jaw and the tongue. Fig. 2 shows the procedure conducted on the left side of the animal. Blunt dissection was performed between the digastric muscle and the left masseter muscle to reveal a tight space within the inner side of the jaw where the inferior alveolar nerve (IAN) can be located (Fig. 2C). Hook retractors were used to maintain surgical access. Using the IAN as a

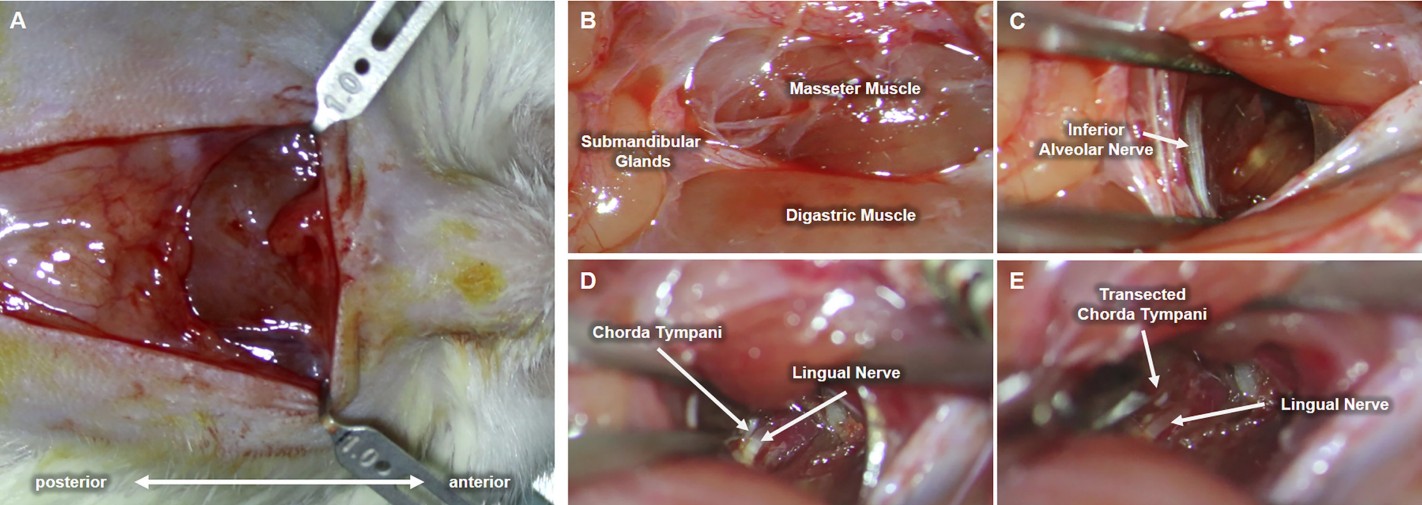

**Figure 2 Procedure for chorda tympani transection.** An incision is made along the midline of the ventral region of the animal's jaw (A). Underlying tissues such as the masseter muscle, digastric muscle, and submandibular glands are located (B). The inferior alveolar nerve (IAN) is then carefully exposed by blunt dissection of the space between the masseter and digastric muscle (C). The lingual nerve is then located deeper within the mediolateral mandibular space used as a tangent landmark (D). Chorda tympani is then traced as an immediate bifurcation of the lingual nerve within the deeper mediolateral mandibular space. The chorda tympani nerve is transected without damaging the lingual nerve (E).

landmark, a deeper anterolateral dissection was done until the lingual nerve was seen. The lingual nerve was then traced dorsally (deeper into the tissue space) until the anterolateral branch of the chorda tympani was identified (Fig. 2D) The chorda tympani was transected using micro-iris scissors by creating a very short segmental cut proximal to the branching from the lingual nerve. The same procedure was then conducted on the opposite side of the animal after which the retractors were removed, surrounding tissues were repositioned, and any excessive bleeding was suppressed using sterile surgical gauze until hemostasis was achieved. The incision was then sutured using a 5-0 monofilament suture followed by disinfection using a povidone-iodine solution. A similar procedure was performed for sham-operated animals just up to the extent of blunt dissection after which the incision was promptly closed. The animals were allowed to recover with food and water before taste preference testing. For clear discussion, the sham-operated group will now be referred to as SHAM and the chorda tympani transected group will be referred to as CTX.

## Microscopy of taste buds

To determine the effect of chorda tympani transection on the innervation, morphology, and expression of taste receptors on the taste buds, a total of 12 animals (CTX = 6 and HAM = 6) were sacrificed using carbon dioxide inhalation overdose at 7, 14, and 28 days post transection. The anterior two-thirds of the tongue was cut from each animal and frozen in preparation for either cryosectioning or protein and gene expression analysis. Cryosections were prepared by embedding frozen tongue tissue samples ($n$ = 3 per category per time point) in optimal cutting temperature compound and cutting 10 um thick sections using a cryostat (Leica CM 1860, Leica Biosystems, Wetzlar, Germany). Prior to fluorescent staining, tissue sections were immersed in 4% paraformaldehyde for

fixation, permeabilized using 0.1% Triton-X100 in PBS, and blocked using 3% bovine serum albumin. The sections were then incubated with rabbit anti-cytokeratin 8 (CK8) antibody (AB59400; Abcam, Cambridge, UK) diluted at 1:1000 and chicken anti-neurofilament H (NF) antibody (AB5539; Millipore, Burlington, MA, USA) diluted at 1:500 in preparation for secondary fluorescent antibody tagging using Alexa Fluor 555 donkey anti-rabbit IgG (A31572; Invitrogen, Waltham, MA, USA) and Alexa Fluor 488 goat-anti chicken IgY (A11039; Life Technologies, Carlsbad, CA, USA), respectively, and was followed by nuclear counterstaining using DAPI. Fluorescently stained sections were then viewed, and micrographs were taken at relevant magnification using Fluoview FV 3000 (Olympus, Tokyo, Japan).

## Protein expression analyses of taste receptors

Three samples of extracted tongue tissues from each categorical group at the aforementioned endpoints were homogenized in preparation for protein and RNA isolation for western blotting and RT-PCR analysis. Using TRIzol® Reagent (ThermoFisher Scientific Co., Waltham, MA, USA), protein and RNA were extracted from tongue tissue samples based on performing sequential precipitation. The protein concentration from extracted tissues was quantified using a protein assay kit. Equal amounts of protein extract were separated using SDS PAGE which was then transferred to a polyvinylidene difluoride membrane. The transfer membrane was then blocked using 5% nonfat milk and was successively incubated with primary antibodies for taste receptor type 1 member 2 and 3 (T1R2 and T1R3), Sonic hedgehog (SHH), gustducin alpha-3 chain (GNAT3), and transient receptor potential melastatin 5 (TRPM5) at 4 °C overnight followed by incubation with respective secondary antibodies at room temperature. The blots were then washed with tris-buffered saline with 0.1% Tween and visualized using electrochemiluminescence reagent. Quantification of the intensities of the visualized blots using ChemiDoc MP System (Bio-Rad Laboratories, Hercules, CA, USA). All readings were normalized using β-actin as a reference.

## Statistical analysis

All data and values are reported as mean with respective standard deviation. Data values were compiled in GraphPad Prism version 8.4.3 for Windows (GraphPad Software, San Diego, CA, USA). There were no criteria for the inclusion/exclusion of animals during the experiment aside from maintaining the aforementioned grouping. Analyses were carried out using the Brown-Forsythe test to determine homogeneity of variance and Welch ANOVA tests followed by a Dunnet T3 *post hoc* test for multiple comparisons of the taste preference gathered from six rats from each categorized group (SHAM/CTX) paired per observation period (7 days/14 days/28 days). Relative protein expression from the western blot was gathered from three gel plots for each categorized group per observation point and was analyzed using unpaired t-tests with or without Welch's correction for unequal variance. The $p$-value $\leq 0.05$ were considered significant for all statistical analyses. Data are available within the article and the Supplemental Materials. Effect size between paired

 

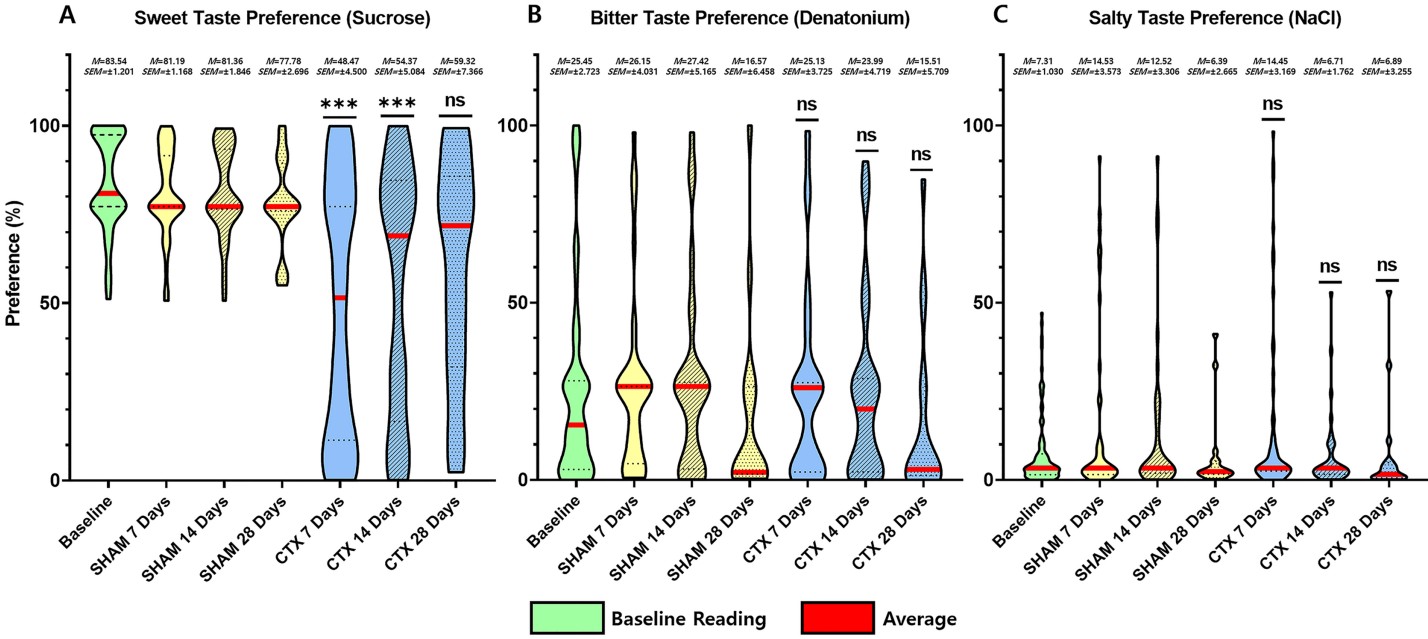

**Figure 3 Behavior preference for sweet, salty, and bitter tastes of CTX and SHAM group at 7, 14, and 28 days after chorda tympani transection.** Results of the behavioral test based on the two-bottle preference tests. Based on the data obtained from the lickometer tests, among the three taste qualities, the preference of the CTX group for sweet taste (A) was significantly altered 7 days after the procedure (Fig. 3A, *p* value = 0.1409) after the procedure. Preference for bitter taste (B) and salty taste (C) remained unaltered after chorda tympani transection (*** = *p* value < 0.001, ns = non-significant, M = mean, SEM = standard error of mean).

SHAM and CTX groups based on respective observation periods were also calculated was on Cohen's d formula.

## RESULTS

### Alteration of taste behavior post Chorda tympani transection

For this study, an animal model was used to determine the effect of chorda tympani transection on the perception of sweet, salty, and bitter taste quality. A total of 36 rats were used, half of which underwent bilateral chorda tympani transection while the remaining half was designated as the sham/control group. All animals underwent behavioral taste testing to establish a baseline reference for taste preference for sweet, salty, and bitter tastes before the surgical procedures. It should be noted that formulations of the tastant solutions are based on minimum concentrations for preference or aversion. Figure 3 shows the violin plot for each taste quality before and after the sham operation or chorda tympani transection. Results indicate that among the three taste qualities, preference for sucrose has been drastically modified when tested 7 days in the CTX group compared to SHAM (Fig. 3A, *p*-value < 0.0001). This change in preference in the CTX groups was further expressed when testing taste preference at 14 days (*p*-value < 0.0001) but has somehow recovered by 28 days after the operation (Fig. 3A, *p*-value = 0.0754). While the mean values of sweet taste preference of the CTX group showed a recovering trend, the data indicated a significantly wider distribution compared to the sham group. Effect size between the SHAM and CTX groups at 7 and 14, and 28 days were calculated to be a 1.217 and 1.0315,

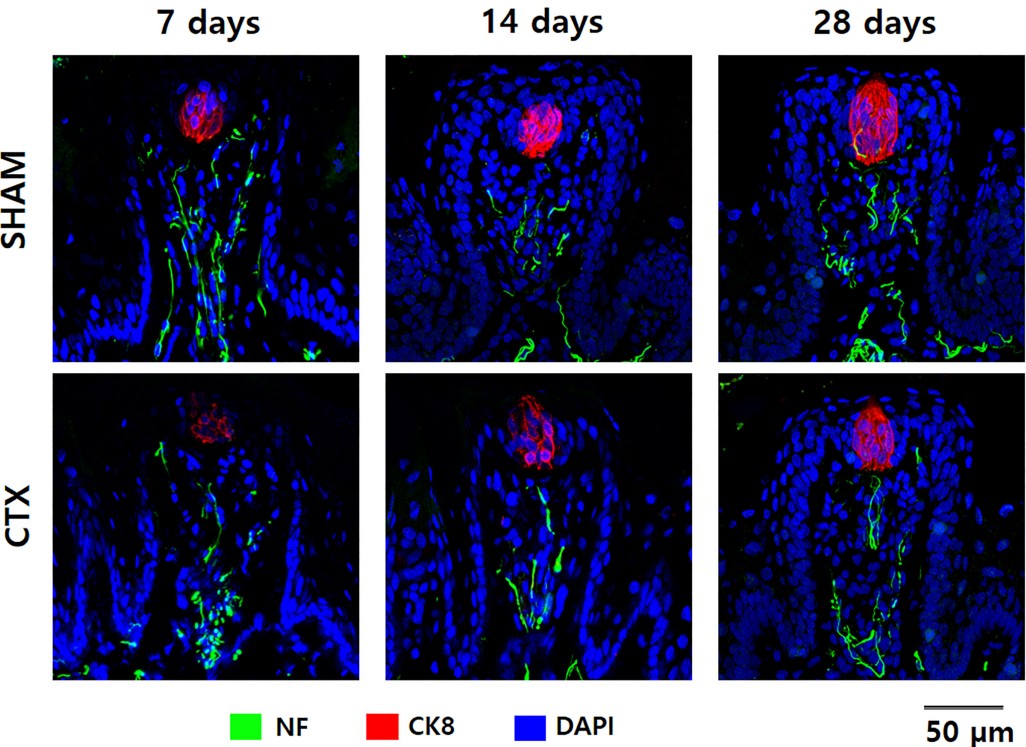

**Figure 4 Immunofluorescent staining of taste buds from SHAM and CTX rat tounge.** Representative confocal micrographs of fungiform papilla lining the anterior portion of the rat tongue stained for neurofilament (NF), cytokeratin 8 (CK8), and nucleus (DAPI).

respectively. As for the bitter (Fig. 3B) and salty (Fig. 3C) taste preferences, pair wise comparison conducted through the *post-hoc* test resulted to no significant difference between SHAM and CTX groups in all specified observation periods.

## Immunostaining for mature taste cells and innervation

To validate the effect of chorda tympani transection on the taste buds, cryosections of the tongue tissue extracted after 7, 14, and 28 days post-transection were immunofluorescent stained for cytokeratin 8 and neurofilament (Fig. 4). Cytokeratin 8 was used to identify mature taste cells within taste buds while neurofilament tagged the innervation for each tastebud. Examination of the micrographs taken from confocal microscopy indicates a significant reduction of CK8 and NF in the CTX group at 7 days evidenced by the reduced fluorescent signature and total morphology relative to the SHAM samples. The fluorescence signature of both CK8 and NF gradually improved after 14 and 28 days post-transection. Both the CTX and the SHAM group showed gradual changes in taste bud morphology when comparing the taste bud shape and size across three observation points. Taste buds in the SHAM group appear smaller and more circular at 7 days when compared to 28 days which appear larger and more elongated. Taste buds in CTX samples at 7 days post transection show a similar circular shape but with a marked reduction of CK8 expression and slight disruption of NF when compared to taste buds from the SHAM

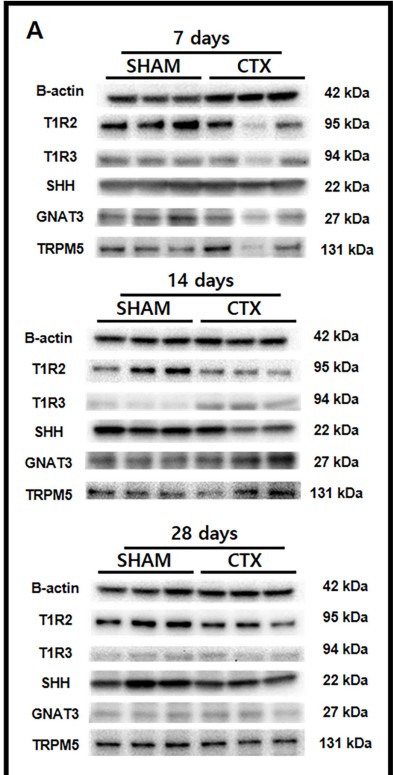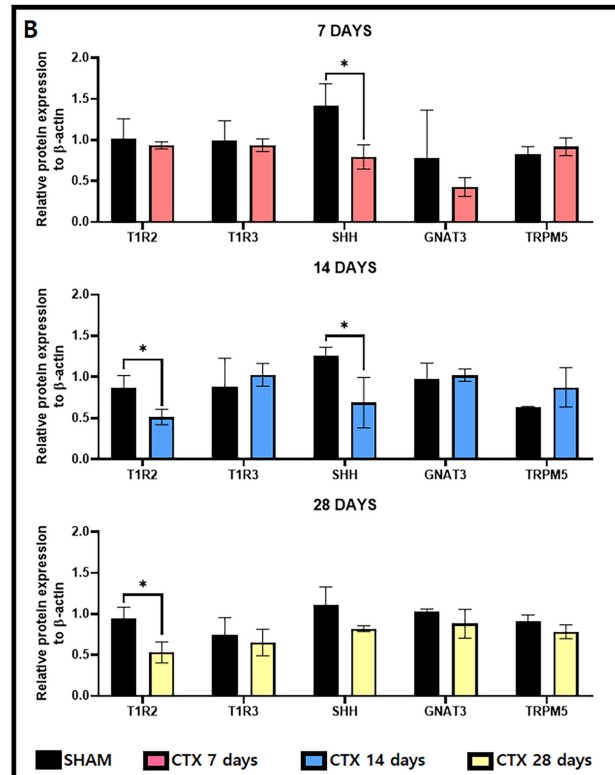

**Figure 5 Western blot images and the relative protein expression normalized to β-actin of taste receptor and transduction proteins.** Representative western blot images (A) and the relative protein expression normalized to β-actin (B). Analysis shows that only SHH (*p*-value = 0.0222) had a reduced expression 7 days after chorda tympani transection. Expression of T1R2 (*p*-value = 0.0218) has also been significantly lowered together with SHH (*p*-value = 0.0373). Twenty-eight days after transection only T1R1 (*p*-value = 0.0179) showed significant reduction while all other proteins appear to match in expression between SHAM and CTX groups (* = *p* value < 0.05).

group at the same time point. Elongation and loose cellular arrangement of the taste bud appears to be more evident at 14 days in CTX group compared to the SHAM group. Taste buds in the SHAM group appear to be more compact and larger than CTX group 28 days after the surgery.

## Expression of taste receptor and transduction proteins

A survey of taste receptor proteins was also conducted to compare the CTX and SHAM groups up to 28 days after chorda tympani transection. Fig. 5A shows the western blot bands generated from protein extracts from SHAM and CTX animals at 7, 14, and 28 days after chorda tympani transection (*n* = 3). Analysis of the quantified band intensities shows that only SHH (*p*-value = 0.0222) was significantly reduced a week after chorda tympani transection. GNAT3 (*p*-value = 0.3646) expression appeared to be also lowered in the CTX group compared to the SHAM group but did not yield as statistically significant. Analysis of the protein expression 2 weeks after chorda tympani transection revealed that T1R2 (*p*-value = 0.0218) have also been lowered together with SHH (*p*-value = 0.0373) while all

other proteins such as T1R3 ($p$-value = 0.5335), GNAT3 ($p$-value = 0.7325), and TRPM5 ($p$-value = 0.2200) showed no significant difference between the SHAM group and the CTX group. This trend was continued towards examination of protein expression twenty-eight days after transection in which both T1R2 and SHH ($p$-value = 0.1498) showed lowered relative expression but only T1R2 ($p$-value = 0.0179) was significantly reduced while all other proteins appear to match in expression between SHAM and CTX groups. Expression of TRPM5 was relatively inconsistent in the CTX group at 7 and 14 days but was not significantly different from the sham group across all observation periods. Among all the proteins surveyed, only T1R2 and SHH showed persistent lowered expression starting from 14 days up to 28 days post transection.

## DISCUSSION

Several studies have been done to determine the effect of CT transection to taste perception. These studies have aided in establishing key points such as age-related plasticity of the CT nerve and its collaterals (*Reddaway et al., 2012*; *Corson & Hill, 2011*); the effect of denervation on the anatomical landscape of the tongue (*Li et al., 2015*; *Sollars, 2005*; *Sollars & Bernstein, 2000*; *Sollars, Smith & Hill, 2002*); and corroborated the importance of CT nerve in instituting taste discrimination (*Pittman et al., 2007*; *Stratford, Curtis & Contreras, 2006*; *Martin & Sollars, 2015*; *Kopka, Geran & Spector, 2000*; *Golden et al., 2011*; *Spector & Grill, 1992*). However, the majority of the observations have been focused on the effect of anatomical changes but have a minimal biomolecular basis for altered taste preference.

This study focuses on the effect of CT nerve transection on the gene and protein expression of taste receptors in the tongue. In this study, three taste qualities were selected tested based on previously surveyed species-specific preferences (*Tordoff, Alarcon & Lawler, 2008*). The sweet taste was selected as a preferred taste quality with no known maximum aversive concentration. The bitter was selected mainly as an aversive taste quality with no known minimal preferred concentration. The salty taste was selected as a middle ground taste quality with known concentrations for aversion and preference. Previous studies on different types of taste cells have shown that both sweet and bitter taste qualities are detected by type 2 taste cells *via* G protein-coupled receptors (GPCR) (*Hoon et al., 1999*; *Adler et al., 2000*; *Zhang et al., 2003*; *Chandrashekar et al., 2000*; *Zhao et al., 2003*). Carbohydrates and sweet-tasting proteins have been found to interact with several binding sites of GPCR T1R2/T1R3 for the perception of the sweet taste quality (*DuBois, 2016*; *Banik & Medler, 2021*). However, bitter taste is primarily associated with GPCR T2R for detection (*Zhang et al., 2003*; *Wong, Gannon & Margolskee, 1996*). Salt taste detection for sodium chloride is based on the presence of ions that could interact with epithelial sodium channels ENac and amiloride-sensitive cells (for preferred low concentrations) (*Vandenbeuch, Clapp & Kinnamon, 2008*; *Menon & Chen, 2019*; *Roitman & Bernstein, 1999*; *Chandrashekar et al., 2010*) or amiloride-insensitive cells (for aversive high concentrations) (*Lu, Breza & Contreras, 2016*; *Roebber, Roper & Chaudhari, 2019*; *Gannon & Contreras, 1995*).

The current study has been designed to determine a biomolecular basis for a shift in taste preference through the analysis of the expression of relevant taste receptor proteins as an effect of CT transection. This was carried out by pairing the protein receptor analysis with a time-matched behavioral taste preference test. Interpretation of change in taste perception is based on the shift of behavioral preference since all taste solutions were formulated at minimum concentrations for detection relevant to taste preference.

The short-term two-bottle test was designed to establish the preference of the test animal whilst minimizing the potential post-oral effect of ingesting large amounts of taste solutions (*Sclafani, 1988*) or possible conditioned preference due to prolonged exposure. (*Kimbrough & Houpt, 2019*) Thus, any change in preference in the CTX group compared to the sham group could indicate an alteration of perception to a specific tastant brought about by the change in expression of taste receptors after denervation.

Both the salty and bitter taste preference showed no change over the 28 days post-transection. Even though several studies have noted the effect of CT transection on salt taste perception (*Kopka, Geran & Spector, 2000*; *Golden et al., 2011*; *Spector & Grill, 1992*), the current results do not corroborate previous findings (*Krimm et al., 1987*) in which sucrose reception was not affected. The method of measuring the behavioral response to taste can be a possible source of discrepancy between the previous (*Krimm et al., 1987*; *Geran & Travers, 2011*) and the current study. Other studies conducted a brief-access test (*Krimm et al., 1987*) while another conducted a lickometer test with multiple concentrations. The current study is based on a combined brief-access test and two-bottle test operated within the context of minimal concentration of preferred or avoided taste quality. The responsiveness of the animals to the minimal aversive concentrations of NaCl and denatonium could be explained by the remaining innervations. Previous research found that the glossopharyngeal nerve (*Geran & Travers, 2011*), which innervates the posterior tongue is sensitive to bitter taste. In addition, both the superficial petrosal nerve (*Sollars & Hill, 1998*) and glossopharyngeal nerve (*Danilova & Hellekant, 2003*) were also found to be responsive to high concentrations of salt. Since NaCl was designated at a minimal concentration for aversion, which is well above the appetitive range, the animal was still responsive to this concentration. While the results of the behavioral preference test for satly and bitter taste quality may be perceived as a "floor effect", this is only because these taste qualities were tested at a single concentration with a known response-aversion. And although the results of the behavioral tests may also indicate that the NaCl concentration tested in this study was more aversive than that of the denatonium, it should be noted that these taste qualities were not tested against each other. Thus a direct comparison between these taste quality would not be suitable under the current circumstances.

Among the three taste qualities tested, only the sweet taste preference is significantly affected by the CT nerve transection. The current results indicated that the CTX group had difficulty discerning between pure water and the lowest preferred concentration of sucrose. Further examination of the behavioral data shows that the sweet taste preference shows gradual renewal (Fig. 3) possibly due to the recovery of enough taste buds to enable the detection of sweet taste at its known lowest preference concentration for the animal strain.

These findings are further supported by the results of the taste bud immunofluorescent staining and protein expression analysis in which expression of T1R3 is relatively unchanged and the morphology of the taste buds observed in the CTX group have been shown to correspond to similar developmental changes, albeit trailing behind, with that of sampled taste buds from the SHAM group. The drastic loss of taste bud volume, visibly confirmed by CK8 positive staining taste cells (Fig. 4), in the CTX group up to 2 weeks after the procedure showed gradual improvement over the course of 28 days. These observations correspond to previous observations on microanatomical changes wherein both papilla density and volume changed after CT transection (*Li et al., 2015*; *Sollars, 2005*; *Sollars & Bernstein, 2000*). It should be noted that although previous research showed that interruption of the chorda tympani nerve results in severely degenerated taste buds over time (*Guagliardo & Hill, 2007*; *Oakley et al., 1993*), approximately 70% of residual taste buds (categorized as atrophic or remnant) in rats can still be present up to three weeks post denervation (*Oakley et al., 1993*). In addition, combined transection of chorda tympani and lingual nerve results to a more severe papilla and taste bud degeneration while interruption of the chorda tympani alone actually allowed for recovery of taste buds after some time (*St John, Markison & Spector, 1995*; *Segerstad, Hellekant & Farbman, 1989*). This suggests that non-gustatory collateral such as the lingual nerve could maintain and support remaining taste buds and fungiform papilla albeit to a limited extent (*Segerstad, Hellekant & Farbman, 1989*). Thus, changes in papillary structure and microanatomy could directly affect the taste detection sensitivity which would consequently alter the level of preference when presented with tastant concentrations at preference thresholds.

Considering that the behavioral tests only yielded significant results upon testing the sweet taste, it was determined that receptor proteins for this taste quality became the focus of observing the protein expression. The survey of the sweet taste receptors (T1R2 and T1R3) and some transduction proteins (GNAT3 and TRMP5) support the change in sweet taste perception brought about by the CT nerve transection. Expression of T1R2 showed consistently lowered protein expression in the CTX group compared to the SHAM group until the 28th day of observation. On the other hand, the T1R3 taste receptor, GNAT3, and TRPM5 taste transduction proteins were relatively unchanged in the CTX group compared to the SHAM group across all observation periods. Considering that T1R2 and T1R3 are coupled G-protein receptors for sweet taste detection while GNAT3 and TRPM5 are the downstream components for basic taste signaling. It follows that disruptions in expression levels in any of these proteins could result in altered sweet taste perception potentially altering preference. Accordingly, any recovery of protein expression related to either receptors or downstream signal components would positively affect re-sensitization for the potentially recover established preference for the given taste quality. The fact that T1R3 is primarily receptive to binding with sucrose while T1R2 effectively binds to several sweeteners (*DuBois, 2016*; *Zukerman et al., 2009*; *Tinti & Nofre, 1991*), thus maintenance of T1R3 expression could aid in adaptive re-sensitization to sucrose solution which would show on the taste preference testing. Sonic hedgehog, a protein that has been found to a play key role in the maintenance of taste buds and taste sensation (*Mistretta & Kumari, 2019*; *Castillo-Azofeifa et al., 2017*; *Ermilov et al., 2016*; *Mistretta & Kumari, 2017*), was also

observed about the possible effect of chorda tympani transection to taste bud development and maintenance. Lowered expression of SHH was evident in the CTX group throughout the observation periods. This noticeable change in SHH protein expression likewise coincides with the findings in the confocal micrographs in which the taste bud in the CTX has a prominent size reduction compared to the SHAM group across all observed time points. However, despite consistently observing lowered levels of SHH and T1R2 protein, the CTX group was still capable of recovering taste buds and sweet taste preference. Reduced SSH expression could result in further microanatomical and biomolecular changes but are unfortunately beyond the scope of the current investigation.

The current study reveals that the changes elicited by the chorda tympani transection are not limited to the micro-anatomical changes in the tongue papilla and terminal field volume of the NST as evidenced by the analyses of taste receptor and taste transduction protein expression. This is the first account of establishing a possible biomolecular basis for the altered taste preference brought about by CT transection in relation to the expression of taste receptor proteins. The analyses of the protein expression for both the receptors (T1R2/T1R3) and the signaling molecules (GNAT3 and TRPM5) parallel the observable recovery for sweet taste preference indicative of functional regeneration or possible compensatory mechanism by the remaining collateral glossopharyngeal nerve. The current results somewhat follow the observations of diminished preference for sweet-tasting compounds in mice that underwent bilateral CT transection (*Danilova & Hellekant, 2003*). Previous studies regarding T1R2/T1R3 have revealed that these receptors can also be found in the gut which enables responsiveness to sweeteners leading to secretion of gut hormones (*Margolskee et al., 2007*; *Sclafani, 2007*). Although it cannot be fully supported by the current findings, it is also possible that the rebound in sweet taste behavior observed in this study might be due to post-ingestive effects as reported by other papers (*Sclafani, 2001*; *Spector, 2015*). Chorda tympani has also been suggested to potentially operate as a discriminator of nutritional information effectively modulating preference while the glossopharyngeal nerve would be responsible for the remaining taste information related to osmotic regulation and aversive tastes (*Tabuchi et al., 1996*). Hence, these findings support the current observations in which bilateral transection primarily affected sweet taste preference (as a preferred taste) with no apparent effect on avoided tastes such as bitter and salty. Although considered a key component for taste signaling, earlier studies have noted the age-related plasticity of the chorda-tympani and its co-laterals. Severance of the CT nerve in rats at P5 resulted did not permit regeneration resulting in altered anatomical development while a similar procedure conducted on adult rats eventually resulted in functional regeneration (*Sollars, 2005*; *Sollars & Bernstein, 2000*; *Martin et al., 2019*).

Nevertheless, it should be noted that the current investigation was limited by the following: it was performed at the animal age at which neural plasticity and functional regeneration can already be achieved, the number of animals was quite limited, and the taste qualities tested at threshold concentration for preference were limited to three. Several factors might also affect the results when testing the behavioral aspect of taste preference and sensitivity in animal test subjects. Aside from the difference in strain,

housing conditions, sex, age, and testing method, the initial introduction of taste components during the development phase of the animal has been found to affect neural development (*Pittman & Contreras, 2002*). Thus the authors would like to interpret these observations with care and consideration. Further detailed investigations regarding umami taste and T1R3 (as a shared taste receptor protein) sucrose re-sensitization would also be needed to verify and elucidate the process of taste preference recovery following CT transection.

## CONCLUSIONS

In conclusion, this study demonstrates the possible correlation between the altered expression of taste receptor/taste signaling proteins and the change in sweet taste preference following chorda tympani transection. The change in sweet taste preference post-denervation can be associated with the disruption in the expression of sweet taste receptors aside from mere micro-anatomical changes in the tongue tissue. Despite the sustained downregulation of T1R2 expression, gradual recovery of sweet taste preference was observed most likely *via* the improved protein expression of T1R3 and GNAT3. Preference recovery, as observed in the behavioral test, is possible in denervated animals although the exact bio-molecular mechanism for the process is yet to be fully elucidated.

### Funding

This work was supported by the grants of the Dankook Institute of Medicine & Optics (DIMO) in 2022. This work was supported by the Korea Medical Device Development Fund grant funded by the Korean government (the Ministry of Science and ICT, the Ministry of Trade, Industry and Energy, the Ministry of Health & Welfare, the Ministry of Food and Drug Safety) [KMDF_PR_20200901_0027-1711137949], the Korea Health Technology R&D Project through the Korea Health Industry Development Institute (KHIDI) funded by the Ministry of Health & Welfare-Republic of Korea [HI20C2088], the Basic Science Research Program backed by National Research Foundation of Korea (NRF) funded by the Ministry of Education [NRF-2020R1I1A3072797, NRF-2020R1A6A1A03043283], and the Leading Foreign Research Institute Recruitment Program through the National Research Foundation of Korea (NRF) funded by the Ministry of Science and ICT [NRF-2018K1A4A3A02060572]. The funders had no role in study design, data collection and analysis, decision to publish, or preparation of the manuscript.

### Grant Disclosures

The following grant information was disclosed by the authors:
Dankook Institute of Medicine & Optics (DIMO).
Korea Medical Device Development Fund: KMDF_PR_20200901_0027-1711137949.
Korea Health Industry Development Institute (KHIDI): HI20C2088.

National Research Foundation of Korea (NRF): NRF-2020R1I1A3072797, NRF-2020R1A6A1A03043283 and NRF-2018K1A4A3A02060572.

## Competing Interests

The authors declare that they have no competing interests.

## Author Contributions

- Andrew Padalhin conceived and designed the experiments, performed the experiments, analyzed the data, prepared figures and/or tables, and approved the final draft.
- Celine Abueva conceived and designed the experiments, analyzed the data, prepared figures and/or tables, and approved the final draft.
- So Young Park performed the experiments, prepared figures and/or tables, and approved the final draft.
- Hyun Seok Ryu performed the experiments, analyzed the data, prepared figures and/or tables, and approved the final draft.
- Hayoung Lee performed the experiments, prepared figures and/or tables, and approved the final draft.
- Jae Il Kim conceived and designed the experiments, authored or reviewed drafts of the article, and approved the final draft.
- Phil-Sang Chung conceived and designed the experiments, authored or reviewed drafts of the article, and approved the final draft.
- Seung Hoon Woo conceived and designed the experiments, analyzed the data, authored or reviewed drafts of the article, and approved the final draft.

## Animal Ethics

The following information was supplied relating to ethical approvals (*i.e.*, approving body and any reference numbers):

Institutional Animal Care and Use Committee at Dankook University.

## Data Availability

The raw data contains the gel plots used for the western blot analyses.

## Supplemental Information

Supplemental information for this article can be found online at http://dx.doi.org/10.7717/peerj.14455#supplemental-information.

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
