# Peer review of "Recovery of sweet taste preference in adult rats following bilateral chorda tympani nerve transection"

_PeerJ, doi:10.7717/peerj.14455_

## Round 0.1 · original submission · Major Revisions

I thank the authors for their submission. I have now received the comments from the reviewers. Please find the comments below. Please respond to each of the comments in detail. I look forward to receiving the revision of the manuscript with the requested changes.

·

Basic reporting

No comment

Experimental design

No comment

Validity of the findings

No comment

Additional comments

This was a nice study that looked at the effect of denervation on the anatomical landscape of the tongue with the biomolecular basis for altered taste preference. The methodology and reporting were both of a very high standard. The sample size was adequate. This study focuses on the effect of CT nerve transaction on the gene and protein expression of taste receptors in the tongue, three taste qualities were selected and tested based on previously surveyed species-specific preferences. However, this study interprets these observations with care and consideration, considering, animal age, the taste qualities tested at threshold concentration for preference, and the development phase of the animal.

Reviewer 2 ·

Basic reporting

-Need for this research have been well emphasized in the introduction section.

-Authors had done assessments of various experimental parameters.

Experimental design

-Authors proposed an innovative and well-designed transparent study with no chances of biases

-Concept of “taste disruption due to downregulation of T1R2 (taste receptor) expression and gradual recovery due to improved protein expression of T1R3 and GNAT3” clearly defined with supporting results

Validity of the findings

-Table and analysis of results are adequate.

-Excellent discussion in relation to previously published studies.

-Conclusion is appropriate with limitations

Reviewer 3 ·

Basic reporting

The manuscript titled "Recovery of sweet taste preference in adult rats following bilateral chorda tympani nerve transection" is clearly written using professional English. It also provides sufficient background and cites references where appropriate.
The few suggestions I have are:
Figure 3 - Please use better resolution and higher font size text. Also, in the figure legend, please explain the annotations 'ns', asterisks (****) etc.,
Figure 5 - mention what * stands for.

Experimental design

The experiments were well designed and the research question well defined stating the knowledge gap.

Validity of the findings

Conclusions were well stated.

Additional comments

This article is well designed and written and would be a good fit for the Journal.
I only have a couple of suggestions for the figures.

Reviewer 4 ·

Basic reporting

In the present manuscript, the authors evaluated changes in behavioral preference testing, immunohistochemistry, and western blot analysis post transection of the Chorda Tympani (CT). They found a suppression in preferences to sucrose at 7- and 14-days post denervation, but no change in preferences to NaCl and Denatonium. They found a decrease in cytokeratin 8 and neurofilament H in denervated animals. Protein expression analyses also found a decrease in Sonic Hedgehog protein expression at 7- and 14-days post denervation and a decrease in T1R2 expression at 14- and 28-days post denervation. While these findings are interesting individually, when put together they don’t make a comprehensive paper and I feel there were a number of major issues in the manuscript.

Experimental design

The rat CT nerve is highly sensitive to sodium and ammonium salts. Sweet and bitter stimuli are weakly effective in the rat CT. Substantial nerve physiology and behavioral discrimination occurs around 30mM. The rat CT is only moderately responsive to 0.5M sucrose and 20mM quinine...the authors rationale for studying low concentrations of sweet taste in the rat CT is unfounded. 100mM NaCl and NH4Cl have been standard stimuli for electrophysiological and behavioral discrimination performance. The authors used 0.562M NaCl, which is strong enough to activate gustatory neurons that innervate other taste papillae in the oral cavity. Regardless, deafferenting the CT and then correlating changes in sweet receptor expression with behavior is unfounded, as the fungiform taste buds are incapable of sending taste information to the brain--as the inputs to the brain from the CT are destroyed.

The motivating operant for behavioral testing was inconsistent across stimuli. Rats were water deprived for sucrose testing, but were water and food deprived for subsequent NaCl and Denatonium testing. The authors justified this choice by stating that it was to increase motivation to drink aversive stimuli (line 139-142). However, it is my understanding that food deprivation has not been shown to affect salt and bitter consumption and it is difficult to interpret preference scores to sucrose relative to those for NaCl and Denatonium given that they were tested under different deprivation conditions. Additionally, food pellets were provided during testing to further motivate animals (line 146), but noncontingent access to food in the middle of training would not act to motivate animals, nor would it train them drink fluids in the lickometer setting. If food was intended to be used as a reinforcer for fluid consumption, an operant procedure would need to be utilized. Lastly, the authors looked at protein expression for T1R2, T1R3, SHH, GNAT3, and TRPM5, which would relate to behavioral testing of sweet and bitter stimuli, but they did not look at expression related to salt responding, such as a-ENaC (Lossow et al., 2020), and bitter responding, such as T2Rs (Chandrashekar et al., 2000).

Validity of the findings

The most important issue was the interpretation of results, particularly the results of the protein expression. The inference that the decrease in sucrose taste preferences is related to the change in protein expression was unfounded. The authors found that T1R2 expression was significantly lower in CTX rats relative to SHAM control animals at 14- and 28-days post denervation, but not at 7-days post denervation. Rats showed a decreased behavioral preference to sucrose at 7- and 14-days post denervation, but not 28-days. If the suppressed preference to sucrose was associated with expression levels of T1R2, animals should have shown a suppressed preference at 28-days post denervation when they had decreased T1R2 expression. More importantly, while the authors demonstrated changes in protein expression post CT transection for the first time, this change in expression is irrelevant to behavioral changes altogether as the nerve is deafferented. Regardless of changes in the expression of proteins in the fungiform taste bud, taste input from fungiform papillae cannot be communicated to the brainstem without the CT nerve being intact.

The findings and methods were inconsistent with previous published research. Most importantly, was the emphasis the authors placed on sucrose reception when previous research has demonstrated that the sectioning the rat CT does not affect sucrose, whereas sectioning the greater superficial petrosal (GSP) nerve does (Krimm et al., 1987). Additionally, bitter stimuli are detected largely on the posterior tongue which is innervated by the glossopharyngeal (GL) nerve (Geran & Travers, 2011) and high concentrations of salt, like 0.562 M, are detected throughout the whole mouth and therefore stimulates the GSP (Sollars & Hill, 1998) and the GL (Danilova & Hellekant, 2003). Therefore, despite sectioning the CT, the rats in the present study could have still been able to respond to the taste stimuli tested via the GSP and GL nerves. The CT has been demonstrated to be necessary for appetitive amiloride-sensitive NaCl responding in rats (Roitman & Bernstein, 1999; St John et al., 1997). If the authors had tested an appetitive midrange concentration of NaCl to show a difference in preferences, it would have greatly increased the enthusiasm for this project.

In addition to this, the immunohistochemical findings in the present manuscript were inconsistent with published research, which shed doubt on the accuracy of CT denervation. Previous research has already established that denervated tongues have significantly fewer taste pores that control tongues (St John et al., 1995). However, the authors have not replicated this here, and were thus unable to prove the CT was cut. The authors present some images (figure 4) of taste papillae wherein the cytokeratin 8 expression in the taste bud can be seen post denervation, but it did not sufficiently demonstrate that the CT was properly transected. Research from Guagliardo and Hill (2007) showed significant changes in Cytokeratin 8 positive cells in fungiform papillae over the course of 15 days. In fact, almost all taste buds disappear in at least the posterior fungiform papillae by 15 days. The data here show a light level of Cytokeratin 8 expression within this timeframe.

While the western blot analysis of protein expression post CT transection was a novel contribution to the field, it is ultimately irrelevant as the ability for taste bud cells to detect stimuli means nothing when the nerve innervating the cells has been transected. The multiple issues contained within this manuscript have substantially reduced my enthusiasm for the paper.

---

## Round 0.2 · Minor Revisions

Please find the comments of the reviewers attached. I will suggest that you consider all the comments of the reviewer and respond to them appropriately. Thank you.

Reviewer 3 ·

Basic reporting

no comment

Experimental design

no comment

Validity of the findings

no comment

Additional comments

no comment

Reviewer 4 ·

Basic reporting

Figure 3. Plotting the standard deviation may not be the best way to graphically represent the data. After CT denervation, the amount of variability for sucrose taste is astounding. The sucrose findings are the heart of the paper, and the way it is represented don’t make a convincing argument. Although the these visually allow for representation of the distribution, it does cast doubt on the magnitude of the effect, and whether any of these p values would withstand correction (example bonferoni). If the authors graphed this as standard error (same as figure) then it would help to improve the visual representation of the effects. Reporting effect sizes would also help to support the findings. Effect sizes could be calculated by a reader based on the means, SD, and n, but it is best to report them.

The authors justified their chosen concentrations based on a 2008 publication. In the rebuttal letter, the authors stated that 0.562M NaCl was supposed to be a "middle ground" with known concentrations of preference and aversion, as 0.562M was where rats first showed avoidance. However, NaCl here was profoundly aversive. Even more aversive than denatonium and with little variance. This needs to be addressed in the manuscript. Specifically, Lines 347-355: This entire paragraph suggests that NaCl at this hypertonic concentration is minimally aversive (based off of the 2008 study), yet, the preference scores in the current study show that it is profoundly aversive--the most aversive stimulus tested. This must be addressed.

Lines 404-407: The rebound in the behavior to sucrose could be due to many factors. Foremost, mice lacking sweet receptors still show preferences for sucrose, as post ingestive cues are still present. This is why researchers like John Garcia in the 1950s (and ingestive researchers since that time, such as Sclafani, Spector, and Smith to name a few) used non nutrient sweeteners like saccharin. In the present study, cutting the CT nerve reduced preferences to sucrose, but animals could have used the GSP, glossopharyngeal (as stated by authors on lines 415-416), as well as post ingestive cues to drive the behavior. This might help to explain the discrepancies in the behavior with molecular markers. As stated in my last review, molecular changes in the taste bud are not the answer for behavior when the nerve innervating the bud is severed. I think the authors need to strongly consider this as a possibility considering that several anatomical studies have shown, in the same animal model, that the time for nerve regeneration is much longer than what is being suggested here. I would suggest the authors include post ingestive feedback as a possibility, as this is a very well-known mechanism for sucrose intake in mice lacking sweet taste.

Experimental design

No comment

Validity of the findings

The variability in the sucrose behavior, following CT nerve denervation, coupled with a modest decrease in the means suggests the p values would not withstand p value correction. The authors do not need to use Bonferroni, as that is a highly conservative test, but they do need to use a correction procedure. They also need to report the effect sizes.

The changes in the sucrose data could be the result of the learning via post ingestive feedback of the caloric load. Mice lacking sweet taste receptors still show learned preferences for sucrose. This is a very well-known effect and was not discussed in the manuscript, though ironically, they cited a Sclafani--who has contributed immensely to this work. Specific lines related to this were reported above in "Basic reporting."

The hypertonic NaCl concentration produced what is referred to as a "floor effect" where animals found it profoundly aversive. No changes were evident with CT nerve denervation, as the entire taste system, and likely the trigeminal system, were stimulated by this hypertonic concentration. It is rare to see such a powerful avoidance. Yet, the authors referred to this as a minimally aversive stimulus, citing a 2008 article. They clearly did not reproduce those results. It is truly not surprising, since the author's data are more in agreement with nearly 100 years of research on NaCl acceptance and avoidance. Only sodium depleted or sodium restricted rats will consume that concentration of NaCl. Rats on a normal sodium diet will avoid hypertonic NaCl, which was clearly shown here as well. Specific lines related to this were reported above in "Basic reporting."

Additional comments

There are numerous typos in this manuscript. I can’t be certain that I found all of them, but it is imperative that the authors comb through the manuscript.

Line 51: “orosenesory” to “Orosensory”
Line 107: Chorda tympani is two words
Line 130: “tastetants” to “tastant”
Line 172: “tymapni” to “tympani”
Line 202: “tastebuds” is two words
Line 237: “Analsysis” to “Analyses”
Line 273: “tastebud” is two words
Line 328: “(ENac)” to “ENaC”
Line 334 and 337: “tastant solutions” to “taste solutions”
Line 350: “A previous research” to “Previous research” More citations are needed. There are numerous physiological recordings from the glossopharyngeal nerve in rodents, showing the wide variety of stimuli for which it is responsive.
Figure 3 legend. There are numerous typos. A character resembling “^” is present in the word “significantly”
Figure 4. Same here. Immunofluorescent has the “^” character as does “neurofilament”. “tounge” to “tongue”
Figure 5. “^” is present twice in the word “significantly”

---

## Round 0.3 · accepted · Accept

Thank you for the revision. With the reviewers comments addressed, your submission is accepted for publication.

Reviewer 4 ·

Basic reporting

No comment

Experimental design

No comment

Validity of the findings

No comment